# Metallic Study of the Invasive Species *Cronius ruber*—Assessment of Toxic Risk

Thabatha Thorne-Bazarra [1], Enrique Lozano-Bilbao [2], Raül Triay-Portella [3], Arturo Hardisson [1], Soraya Paz [1], Carmen Rubio-Armendariz [1], Verónica Martín [4] and Angel J. Gutiérrez [1,*]

1   Toxicology Area, Universidad de La Laguna, 38071 La Laguna, Spain; tbtabatha@gmail.com (T.T.-B.); atorre@ull.edu.es (A.H.); spazmont@ull.edu.es (S.P.); crubio@ull.edu.es (C.R.-A.)
2   Departamento de Biología Animal y Edafología y Geología, University of La Laguna, Canary Islands, 38071 Tenerife, Spain; lozaenr@gmail.com
3   Grupo en Biodiversidad y Conservación, IU-ECOAQUA, Universidad de Las Palmas de Gran Canaria, ULPGC, 35200 Telde, Spain; raul.triay@ulpgc.es
4   Canary Health Service, 35004 Las Palmas de Gran Canaria, Spain; cmarleov@gobiernodecanarias.org
*   Correspondence: ajguti@ull.edu.es; Tel.: +34-922-318-905

**Abstract:** Three toxic heavy metals (Cd, Pb, and Hg) were analyzed in the newly found invasive species in the Canary Islands, *Cronius ruber*. Its high growth rate and its widely varied diet are affecting the Canary marine ecosystem. The study was conducted using electrothermal vaporization atomic absorption spectrometry (GF-AAS) and cold-vapor atomic absorption spectrophotometry (CV-AAS). Significant differences were found in terms of the location and sex of the specimens, with the highest concentrations being found in areas with higher tourism activity and in the female specimens. On the other hand, the conclusion of the study is that human consumption of this species does not pose any toxic risk to public health, as the levels obtained in muscle tissue do not exceed the established limits for these metals. Therefore, its consumption and the fishing of this species can stop the proliferation of the same in the Canary coasts and thus not be harmful for the ecosystem.

**Keywords:** *Cronius ruber*; metal; human risk; Canary; marine invasive species



## 1. Introduction

Pollution is now a major environmental concern as the world's biodiversity is highly susceptible due to the fragility of aquatic ecosystems, whose local biota has unique ecological communities [1–3]. There are various sources of pollution, both natural and artificial, the latter being produced by human activities [4–6]. The heavy metal pollution of water markedly affects both food safety and public health. Due to its high toxicity, the impact on health caused by the prolonged exposure to or bio-accumulation of heavy metals is a cause of concern [7–9]. Depending on the type of metal or metalloid, conditions ranging from damage to vital organs to their development occur, which may contribute to the degradation of marine environments by reducing the diversity of organisms [10–12]. These heavy metals, compared to other pollutants, are not biodegradable and have a global ecological cycle in which natural waters are the main entry routes, which can result in serious ecological and biological alterations, not only to the ecosystems but also to humans [13,14]. Crustaceans and crabs, in particular, are of much nutritional interest in terms of human nutrition because of their high nutritional value. The muscle (white meat), the hepatopancreas (brown meat), and the gonads are either consumed separately or in a mixture of them [15–17]. These invertebrates are capable of accumulating toxic metals such as Cd and Pb in their organs and muscle tissue. The accumulation of toxic metals in different crab organs and tissues varies depending on several factors, such as species of crab, type of tissue, location, food, sex, and size. Some species of crabs can play an important role in ecology as biomarkers of pollution [18–22].

Human interference, such as the overexploitation of marine resources and the alteration of the coastline, endangers the marine biodiversity of the Canary coasts on a daily basis. With the increase in the rates of anthropic disturbance, world trade, and global climate change, alarms bells have been ringing around the world about the increase in biological invasions in most ecosystems [23,24]. The introduction and spread of nonnative species by humans are considered one of the main threats to biodiversity [25–27]. Consequently, interest in species introduction studies has grown around the world in recent years. Biological invasions into the environment are a growing problem because nonindigenous organisms can affect the structure and functioning of native communities [28–31]. Bioindicator species are biological tools that allow the total or partial evaluation of ecological systems, so they can be considered as estimators of the biodiversity of a system, acting at different hierarchical levels (genes, species, populations, communities, and landscapes) and determining different components of biodiversity. Therefore, the state of the ecosystem can be estimated, with invasive species being good bioindicators [32]. The settlement or passage of nonnative marine species around the Canary Islands has increased over the last thirty years [33,34]. The invasive crab *Cronius ruber* has a mesopredator role where several threatened or commercially important species are the major components of its diet in the region [32]. Therefore, the objective of this study is to reveal if *Cronius ruber* can be a bioindicator of pollution by heavy metals in the muscle and exoskeleton, to determine if they can be used as food, as this crab is an invasive organism, and fishing can decrease its abundance in the ecosystem. For this reason, a risk assessment is carried out for Pb and Cd, determining if it is suitable for human consumption.

## 2. Materials and Methods

First, 60 rowing crab (*Cronius ruber*) specimens were collected in two locations in Gran Canaria (El Pajar and Agaete) in 2019 (Figure 1). The sex, the different biometric measurements, and the metal content were determined in the crabs from both locations. Here, 60 samples were captured, of which both the muscle and exoskeleton were used, 30 samples in El Pajar and 30 samples in Agaete, of which 15 samples were from female specimens and 15 samples from male specimens.

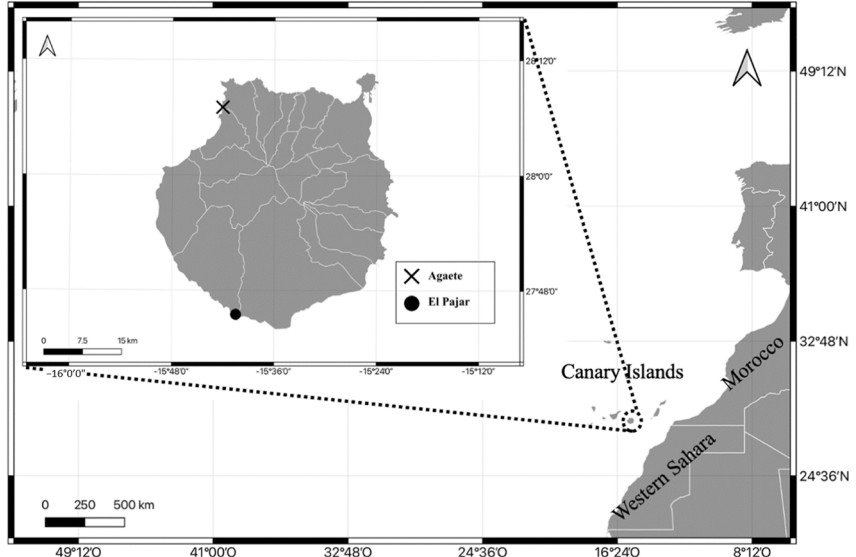

**Figure 1.** Map of the island of Gran Canaria with the localities selected in the study.

The methodology for the preparation of samples and the subsequent determination of metals in them was performed in two different ways depending on the type of sample for analysis. The methodology to determine the metal content in the muscle samples was mineralization by incineration in a muffle furnace and the method for the exoskeleton

samples was wet digestion in a microwave oven. Nutritional and toxicological assessment was carried out following the recommendations of AESAN [35], knowing the Estimated Daily Intake (EDI) considering the reference values of the Recommended Daily Intakes (RDI) provided by FESNAD, and the Admissible Daily Intake (ADI) of EFSA [36].

The nutritional study and the risk assessment were conducted using the following formulas for calculating ADI = NOAEL (No Observed Adverse Effect Level)/F(Intraspecific Factor); EDI = (C.metal (Metal Concentration) × Cons(Consumption))/bw (Body weight); and MOS (Margin of Safety index) = EDI/ADI.

Step 1: The crabs were separated. Step 2: Biometric measurement of the sex, weight, and length was carried out. Step 3: Muscle tissue and exoskeleton tissue were separated. Step 3: The muscle tissue (3 g) samples were placed in the oven for 24 h at a temperature of 80 °C. They were then put in the muffle oven for 24–48 h at a temperature of 450 °C ± 25 °C, until white ashes were obtained. If these were grayish, they were put back in the oven for 24 h after adding pure nitric acid to each one. Subsequently, each of the samples was filtered with a 1.5% nitric acid ($HNO_3$) solution and made up to 25 mL with the same solution (Lozano-Bilbao et al., 2020b). Step 4: Exoskeleton samples (1 g) were processed by means of a microwave digester. To do this, one gram of exoskeleton from each of the samples was used and 2 mL of hydrogen peroxide ($H_2O_2$) and 4 mL of nitric acid ($HNO_3$) were added. After the microwave digester program finished, the samples were passed through 10 mL volumetric flasks and made up to the mark with distilled water until the indicated amount was reached. Step 5: The metal content of the processed *Cronius ruber* samples was determined by using atomic absorption spectrometry with electrothermal vaporization (GF-AAS) with a graphite chamber, and cold-vapor atomic absorption spectrophotometry (CV-AAS) with a flow injection system (FIAS) (Figure 2) [37,38]. A quality control solution was used to assess the determination accuracy every 10 samples. The precision of the analytical procedure was evaluated by analyzing international reference standard materials DORM-1 and TORT-1 (National Research Council of Canada). All data were presented as milligrams per kilogram and wet weight (mg/kg w.w.). Blanks and standard reference materials were run together with samples.

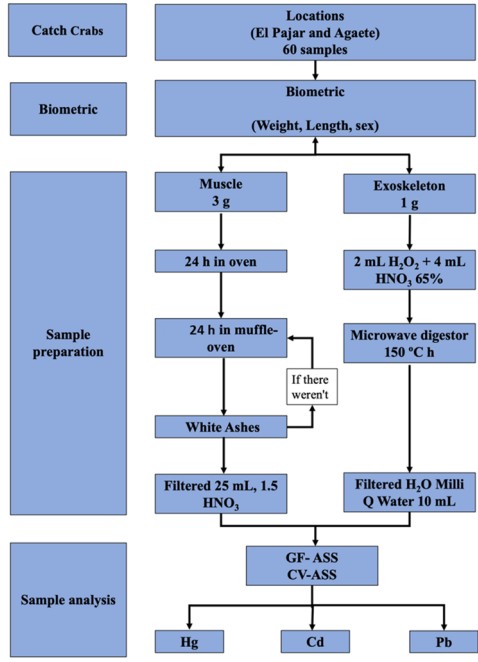

**Figure 2.** Flowchart of the methodology used.

### 3. Statistical Analysis

To determine whether there were variations in the content of heavy metals and trace elements between the analyzed samples, a statistical analysis was performed using a multivariate permutational analysis of variances (PERMANOVA) with Euclidean distances [39]. In all analyses, 9999 interchangeable unit permutations and a posteriori comparisons were used to determine differences between levels of significant factors ($p$-value < 0.05) [40]. The PRIMER 7 and PERMANOVA + v.1.0.1 statistical packages were used for the statistical analyses. To control the variability of the data and the variables, they were treated with a triangular matrix calculated from the Euclidean distances and the square root of all the data.

The variables included in the analysis were the concentration in mg/kg of the heavy metals Cd, Pb, and Hg and biometric data. Two one-way designs were performed with two levels of variation, one for the levels in El Pajar and Agaete and the other for the levels of variation in each type of female and male tissue. A two-way design was performed using the following factors: zone (El Pajar and Agaete) and tissue (exoskeleton and muscle) [41].

### 4. Results

#### 4.1. Biometric

Table 1 shows the biometric data in the two study areas; we found that the females in El Pajar are smaller and have less weight. There are no significant differences in the PERMANOVA analysis between the weight and length of the studied areas.

**Table 1.** Biometric data (sex, length, and weight) of *Cronius ruber* in the study locations.

| Location | Sex | Length (Width) | Weight |
|---|---|---|---|
| Agaete | Male | 78.63 ± 9.91 (78.9–83.4) | 135.06 ± 48.11 (119.9–165.1) |
| | Female | 76.63 ± 5.84 (69.2–87.4) | 101.66 ± 26.93 (51.2–137.2) |
| El Pajar | Male | 75.9 ± 7.61 (57.1–84.6) | 112.93 ± 30.25 (46.6–157.3) |
| | Female | 68.58 ± 9.22 (77.5–84.2) | 120.05 ± 31.95 (119.5–127.5) |
| $p$-value | | 0.342 | 0.209 |

#### 4.2. Metal Concentrations

Table 2 shows the metal concentrations with their respective standard deviation, of both the muscle and exoskeleton samples of the specimens, as well as where they were collected and their sex.

The concentrations of the metals in Table 2 show that the Cd concentration in the muscle is higher in the samples collected in the El Pajar area, with a mean average level of 0.008 ± 0.006 mg/kg in the females. Regarding the exoskeleton samples, there is no variation in the concentrations between males and females from the two study areas. On the other hand, the Pb levels in the muscle tissue are higher in the female specimens from Agaete than the female and male specimens from El Pajar and the males from the same study area. The level of Pb in the exoskeleton is high and is higher than the other concentrations obtained (0.853 ± 1.391 mg/kg) in the Agaete females. Finally, the Hg concentrations are higher in the exoskeleton in Agaete in both sexes than in El Pajar.

**Table 2.** Mean concentration and standard deviations in *Cronius ruber* (mg/kg), according to sex, location, and type of sample.

| | Location | Sample | Mean Concentration (mg/kg) ± SD vs. the Location Factor, and Sample | Sex | Mean Concentration (mg/kg) ± SD vs. the Location Factor, Sample, and Sex |
|---|---|---|---|---|---|
| Cd | El Pajar | Muscle | 0.003 ± 0.002 | Male | 0.002 ± 0.001 |
| | | | | Female | 0.008 ± 0.006 |
| | | Exoskeleton | 0.001 ± 0.001 | Male | 0.001 ± 0.000 |
| | | | | Female | 0.005 ± 0.004 |
| | Agaete | Muscle | 0.003 ± 0.003 | Male | 0.001 ± 0.001 |
| | | | | Female | 0.004 ± 0.003 |
| | | Exoskeleton | 0.004 ± 0.004 | Male | 0.001 ± 0.000 |
| | | | | Female | 0.005 ± 0.004 |
| Pb | El Pajar | Muscle | 0.004 ± 0.003 | Male | 0.004 ± 0.003 |
| | | | | Female | 0.005 ± 0.002 |
| | | Exoskeleton | 0.199 ± 0.096 | Male | 0.178 ± 0.083 |
| | | | | Female | 0.338 ± 0.045 |
| | Agaete | Muscle | 0.012 ± 0.025 | Male | 0.005 ± 0.004 |
| | | | | Female | 0.013 ± 0.027 |
| | | Exoskeleton | 0.766 ± 1.308 | Male | 0.202 ± 0.105 |
| | | | | Female | 0.853 ± 1.391 |
| Hg | El Pajar | Exoskeleton | 0.002 ± 0.003 | Male | 0.001 ± 0.001 |
| | | | | Female | 0.009 ± 0.004 |
| | Agaete | Exoskeleton | 0.009 ± 0.005 | Male | 0.003 ± 0.003 |
| | | | | Female | 0.009 ± 0.005 |

Regarding the PERMANOVA analyses of muscle tissue, Table 3 shows that there are no significant differences as the *p*-values obtained are greater than 0.05. However, in Table 4, one can see a significant difference in Cd in El Pajar, with a value of 0.031. Therefore, there is a relationship in terms of sex and location with Cd. In Table 2, which shows the study of the relationship between sex and locations, one can see a higher Cd concentration in the female specimens from El Pajar of 0.008 ± 0.006 (mg/kg) compared to males from the same location with a value of 0.002 ± 0.001 (mg/kg) (Figure 3). Principal coordinate analysis (PCoA) was performed to verify these results. The first PCoA was performed with muscle tissue and Cd and Pb (Figure 4). The X-axis in Graph 1 represents 82.5% of the total variation and the Y-axis represents 17.5% of the total variation, given that the samples are not separated; therefore, the results obtained in the PERMANOVA show that there are no significant differences between locations (Table 3). This crab has a fast growth rate and a very varied diet, being an omnivorous species that feeds on algae with small invertebrates, without seasonal variations; components 1 and 2 are significant for all analyses, *p*-value < 0.05.

**Table 3.** PERMANOVA analysis of heavy metals in muscle tissue with the location and sex factors.

| Metal | Cd | | Pb | |
|---|---|---|---|---|
| Location | El Pajar | Agaete | El Pajar | Agaete |
| Female vs. Male | 0.031 * | 0.174 | 0.549 | 0.633 |
| El Pajar vs. Agaete | 0.477 | | 0.15 | |

* *p*—significant value < 0.05.

**Table 4.** PERMANOVA analysis of heavy metals in exoskeleton with the location and sex factors.

| Metal | Cd | | Pb | | Hg | |
|---|---|---|---|---|---|---|
| Location | El Pajar | Agaete | El Pajar | Agaete | El Pajar | Agaete |
| Female vs. Male | 0.013 * | 0.373 | 0.072 | 0.52 | 0.011 * | 0.147 |
| El Pajar vs. Agaete | 0.011 * | | 0.087 | | 0.002 * | |

* *p*—significant value < 0.05.

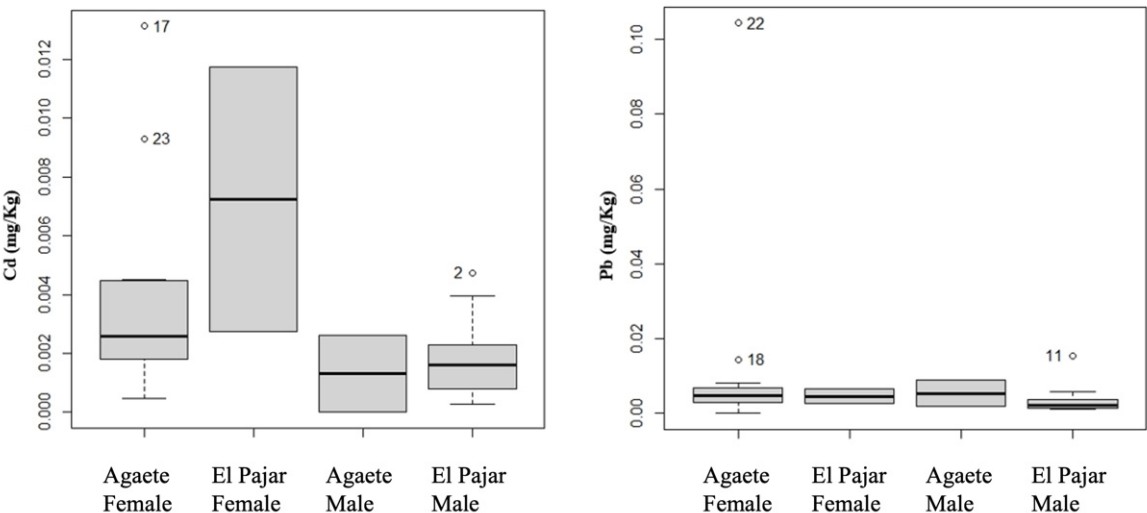

**Figure 3.** Box plot for Cd and Pb (mg/kg) in muscle tissue with the location and sex factors.

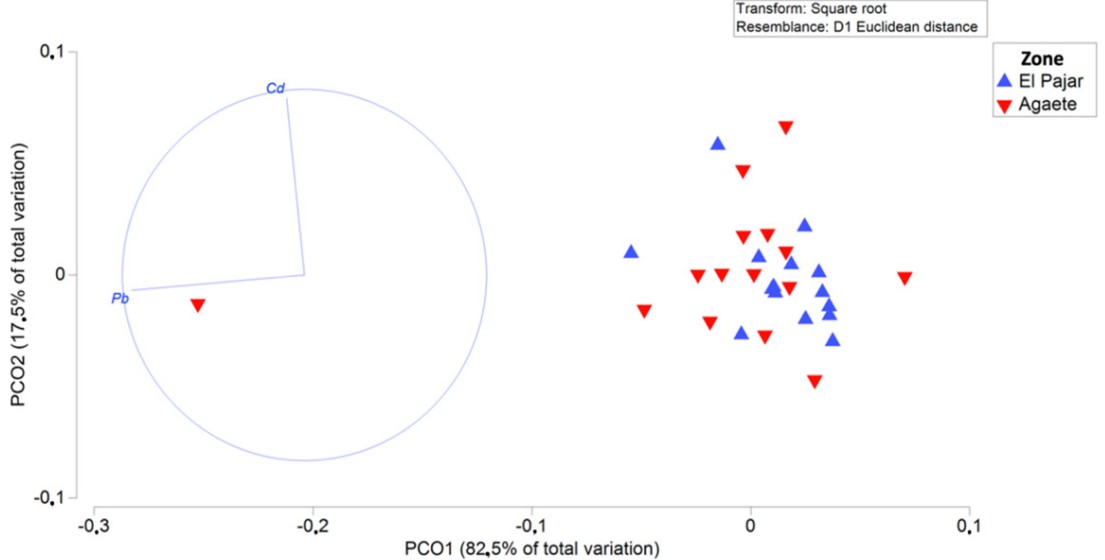

**Figure 4.** Principal Coordinate Analysis (PCoA) of muscle tissue with the location factor.

### 4.3. Statistical Analyses

The PERMANOVA analyses indicate that there are significant differences in the exoskeleton. In Table 4, the *p*-value of Cd is 0.011, while that of Hg is 0.002, less than 0.05 in both cases. In addition, the highest levels of Cd and HG are 0.004 ± 0.004 mg/kg and 0.009 ± 0.005 mg/kg in the Agaete location, respectively.

On the other hand, Table 4 shows significant differences in El Pajar for Cd and Hg, with *p*-values of 0.013 for Cd and 0.011 for Hg, Cd concentrations of 0.001 ± 0.000 mg/kg in males and 0.005 ± 0.004 mg/kg in females, and Hg concentrations of 0.001 ± 0.001 mg/kg in males and 0.009 ± 0.004 mg/kg in females for Hg (Figure 5). This is corroborated with

the PCoA where the X-axis represents 98.8% of the total variation and the Y-axis 0.9% of the total variation (Figure 6).

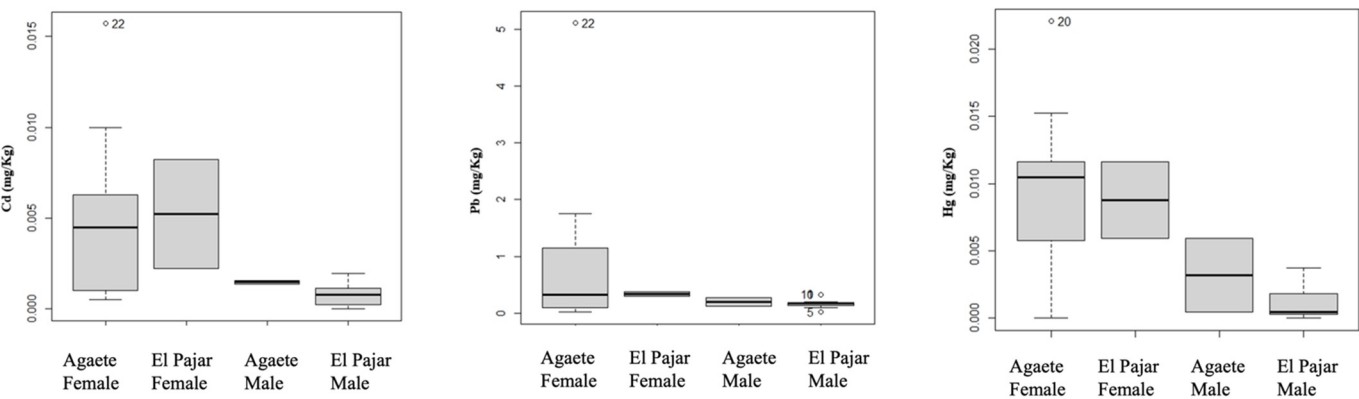

**Figure 5.** Box plot for Cd, Pb, and Hg (mg/kg) in exoskeleton with the location and sex factors.

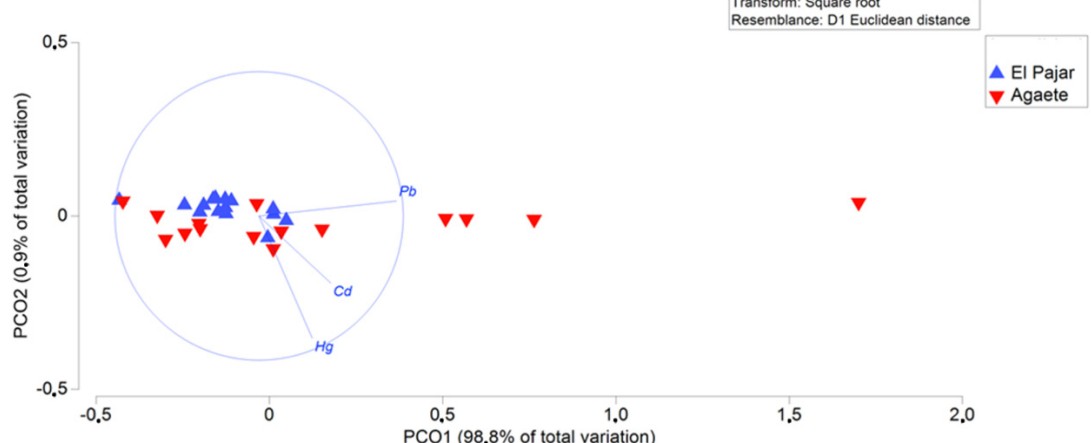

**Figure 6.** Principal Coordinate Analysis (PCoA) of the exoskeleton with the location factor.

The PERMANOVA in Table 5 shows that there are significant differences in Cd and Pb in the two locations. This suggests that there could be marine pollution from the metals studied here in both locations, derived from uncontrolled discharges from industry, urban areas, or agriculture. However, in the case of this type of pollution affecting *Cronius ruber*, although the concentrations of some metals are high, they are not worrying. These concentrations are shown in Table 1, with the mean average concentrations according to location and type of sample. El Pajar concentrations of Cd in muscle tissue and the exoskeleton are $0.003 \pm 0.002$ mg/kg and $0.001 \pm 0.001$ mg/kg, respectively, compared to concentrations in Agaete of $0.003 \pm 0.003$ mg/kg in muscle and $0.004 \pm 0.004$ mg/kg in the exoskeleton. On the other hand, in the case of Pb, the levels are $0.004 \pm 0.003$ mg/kg in muscle and $0.199 \pm 0.096$ mg/kg in the exoskeleton in El Pajar, and $0.012 \pm 0.025$ mg/kg in the muscle tissue and $0.766 \pm 1.308$ mg/kg in the exoskeleton in Agaete (Table 1), and these differences are confirmed by the PCoA that represents the type of sample and the location. The X-axis of this graph represents 99.5% of the total variation and the Y-axis represents 0.5% of the total variation (Figure 7).

**Table 5.** PERMANOVA analysis of heavy metals in muscle tissue and exoskeleton with the location factor and type of sample.

| Location | Cd | | Pb | |
|---|---|---|---|---|
| | El Pajar | Agaete | El Pajar | Agaete |
| Muscle vs. Exoskeleton | 0.001 * | 0.015 * | 0.001 * | 0.001 * |

* *p*—significant value < 0.05.

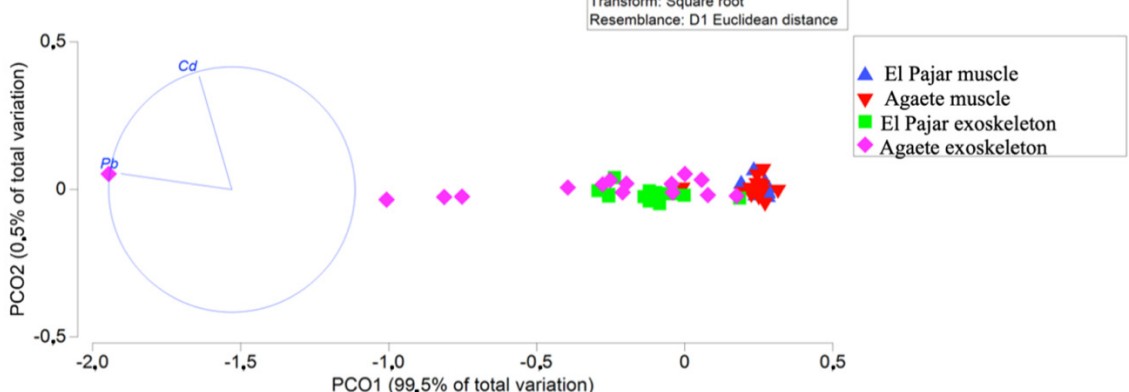

**Figure 7.** Principal Coordinate Analysis (PCoA) of muscle tissue and exoskeleton with the location factor and type of sample.

## 5. Discussion

### 5.1. General Features

Exposure to these metals may be due to the animals being continuously exposed to discharges present in the area where they live, thus causing their own food web, the water column, and sediments to be contaminated. Therefore, there are significant differences in terms of heavy metal pollution in the two study locations. These differences according to location may be due to the tourism in these areas, as a high demand for tourism leads to more urbanization and, consequently, a greater discharge of pollutants into the marine environment [42–44]. Regarding the significant differences concerning the sex factor and the location, females have the highest concentrations, which could be due to higher energy requirements due to the stage of their reproductive cycle [45–48]. In addition, shifts in *Cronius ruber* diets with ontogeny and sex indicate that juveniles and mean-sized crabs, demographically dominated by females, have a diversified diet and consume more prey items per day than dominant male crabs [32]. Furthermore, differences can also be found in the concentrations in muscle tissue and the exoskeleton, with the latter having higher concentrations. This higher concentration of heavy metals in the exoskeleton suggests the animal's ability to sequester heavy metals from the body safely [49–51]. Studies carried out by [52] determined the existence of several factors influencing the elimination of metals from the body of marine animals. They include factors such as time, temperature, interacting agents, age, metabolic activity of the animals, and the biological half-life of metals [53]. Other studies on crabs by [54] determined a detoxification mechanism by which these animals release toxic elements during the molting process. In this respect, a higher concentration of metals could have been found in specimens in the present study because they had not undergone the molting process. With the results obtained, we observe that it is the exoskeleton tissue that presents the highest concentrations of metals, as all crustaceans eliminate metals through molting, because it is a rapid detoxification process; metals such as Cd accumulate in calcareous areas, doing so with the molting, and large concentrations of this metal are eliminated [55–57]. It should be noted that, as in any crustacean, the highest levels of toxic heavy metals would be found in the hepatopancreas [58–60]. The range of expansion of this invasive species is not known; there are currently several groups

investigating them. The Island of Gran Canaria was chosen because it is the first record in the Canary archipelago, so it was estimated that they expanded from of this island [32].

The Cd concentrations found here in the exoskeleton are due to the tendency of this metal to be replaced by calcium [61], as it has similar chemical properties. It has also been reported that this metal causes adverse effects in some crab species, where an inhibitory response has been observed in the respiration rate of these crustaceans [62]. On the other hand, a study conducted by Knowlton [63] determined that a lower concentration of Pb in crabs seems to be more related to the adsorption of this element than to its bioaccumulation. However, the data of the present study showed high concentrations of this metal, so there is bioaccumulation [64]. Furthermore, acute Pb exposure affects crabs by reducing the survival rate of their sperm and producing damaging effects at the cellular level in their testes and accessory glands, which are most likely related to oxidative stress induced by Pb [65]. Regarding Hg concentrations, they may affect multiple metabolic processes in exposed organisms, because, as with Pb, they form organometallic compounds [66]. We observe that in most of the data, the concentrations are higher in Agaete, which may be due to the fact that despite not being a tourist place, it is a busy port area, so pollution through the search makes it very obvious. This is why *Cronius ruber* can be a good bioindicator of metals such as Cd, Pb, and Hg in their tissues for port pollution.

*5.2. Comparison with Other Studies*

Table 6 shows that the levels in all the toxic heavy metals studied (Cd, Pb, and Hg) in the present study are lower than others conducted by different authors; however, the presence of these metals in all the conducted studies is due to uncontrolled discharges from industry and urbanized areas. All studies reporting high levels in the concentration of metals suggest notable problems in the marine pollution of the study area. The study carried out by [67] on *Grapsus adcensionis*, around the island of Tenerife (Canary Islands), reported the highest level of Pb in the exoskeleton, which could have been due to the oil refining plants near the study zone. Although the exoskeleton values of this metal are high, those of the muscle tissue are also high, exceeding the European limits regarding the risk of Pb poisoning when ingesting more than 1762.5 g/week, and for Cd poisoning with an intake of more than 152.57 g/week. On the other hand, the Pb concentration in the muscle of *Chasmagnatus granulata* studied by [68] is also high, and the consumption of these crustaceans is prohibited because their Pb levels are higher than those allowed by legislation for their consumption. The metals studied by [69] in the muscle tissue of *Portunus petagicus* suggest that there is a close relationship between the animals studied and the marine sediment where they live, as they are good indicators for monitoring this type of pollution [70]. Marine organisms, including fish, squid, and crustaceans, can accumulate heavy metals through direct absorption [69]. This may be related to several factors such as habitat, bioavailability of pollutants in the environment, and dietary absorption, as crustaceans feed mainly through sediments, which are considered the main reservoir of pollution [71]. The Pb concentrations in *Scylla serrata* are lower in the two types of samples compared to the other studies, but higher than those of *Cronius ruber* [72]. The Hg levels in studies on *Pseudocarcinus gigas* and *Carcinus* sp. are high when compared to those found in the present study [73,74]. This is a concern as Hg has historically produced significant environmental poisoning episodes. However, the concentrations in the muscle tissue of *Cronius ruber* do not pose any risk to public health. *Callinectes sapidus* can be considered a bioaccumulative and bioindicator organism as with other species of crabs [75]. Although the results found here for *Cronius ruber* suggest that the specimens are suitable for consumption, in addition to human consumption, they can also be used as bioindicators in future studies in order to determine whether an area is contaminated.

**Table 6.** Comparison of the mean concentrations of heavy metals in the present study with others in different parts of the world.

| Species | Sample | Unit. | Cd | Pb | Hg | Study |
|---|---|---|---|---|---|---|
| *Cronius ruber* | Muscle—El Pajar | mg/kg | $0.003 \pm 0.002$ | $0.004 \pm 0.003$ | | The present study |
| | Exoskeleton—El Pajar | | $0.001 \pm 0.001$ | $0.199 \pm 0.096$ | $0.002 \pm 0.003$ | |
| | Muscle—Agaete | | $0.003 \pm 0.003$ | $0.012 \pm 0.025$ | | |
| | Exoskeleton—Agaete | | $0.004 \pm 0.004$ | $0.766 \pm 1.308$ | $0.009 \pm 0.005$ | |
| *Grapsus adcensionis* | Muscle | mg/kg | 1.148 | 1.089 | | [67] |
| | Exoskeleton | | 4.736 | 13.92 | | |
| *Portunus petagicus* | Muscle | $\mu g \cdot g^{-1}$ | 0.08 | 0.008 | | [68] |
| *Chasmagnatus granulata* | Muscle | $\mu g/g$ | | 10.00–13.20 | | |
| *Scylla serrata* | Muscle | $\mu g/g$ | | 0.17–0.21 | | [72] |
| | Exoskeleton | | | 0.15–0.16 | | |
| *Callinectes sapidus* | Muscle | mg/kg | $0.24 \pm 0.05$ | $0.88 \pm 0.04$ | | [75] |
| *Pseudocarcinus gigas* | Muscle | mg/g | 0.05 | | 0.3 | [74] |
| *Carcinus* sp. | Muscle | mg/kg | 0.07 | 5.52 | 0.11 | [73] |

### 5.3. Estimated Daily Intake

The values found in the present study do not pose any risk for consumption and suggest that there would be no toxic risk from the ingestion of this crustacean, *Cronius ruber*. As the part suitable for consumption is only the muscle, the different indices shown in Table 7 were calculated for Cd and Pb. The amount of Cd that can be consumed per week is 2.5 µg/kg. The ADI index for both locations is 0.0001, which is low, so it is not problematic. On the other hand, the MOS index is $7.49 \times 10^{-8}$. Therefore, as this is less than 1, the toxic risk from the ingestion of this species for human health is negligible. The number of kg of *Cronius ruber* from the two study areas that would need to be consumed to pose a risk is as high as 734 kg. This amount would never be consumed in practice, so once again, it is true to say that its consumption is not dangerous to human health.

**Table 7.** Risk assessments (Pb, Cd) for muscle samples, the ADI (Estimated Daily Intake), MoS (Margin of Safety), and the amount in kg of muscle that would have to be consumed every day throughout life for it to be harmful to the human being have been calculated.

| Metal/Limits | Index | El Pajar | Agaete |
|---|---|---|---|
| Cd 2.5 µg/kg/week | ADI | 0.0001 | 0.0001 |
| | MoS | $7.49 \times 10^{-8}$ | $7.49 \times 10^{-8}$ |
| | Kg | 734 | 734 |
| Pb 0.5 µg/kg/day | IDA | 0.087 | 0.025 |
| | MoS | 0.001 | 0.02 |
| | Kg | 356 | 104 |

In the case of Pb, the amount that can be safely consumed per day is 0.5 µg/kg of body weight. The ADI in El Pajar is higher than that in Agaete, being 0.087 and 0.025, respectively. These values do not pose a health risk. Regarding MOS, it is higher in Agaete, with a value of 0.02 compared to El Pajar with a value of 0.001. As in the case of Cd, it does not pose any toxic risk, as it is less than 1. Finally, the number of kg that could be ingested without toxic risk is 356 kg in *Cronius ruber* from El Pajar and 104 kg in those from Agaete. Such quantities would never be consumed in practice. Thus, its consumption could not affect the consumer from the point of view of its toxic heavy metal content.

Furthermore, the presence of these two metals in the muscle of the specimens is not high and, as such, *Cronius ruber* is fit for human consumption, as long as the concentrations of these metals are lower than the maximum levels set by international and national regulatory agencies, with the ADI of these metals at 0.025 mg/kg/day for Cd and 0.3 mg/kg/day

for Pb [76–82] (AECOSAN, 2006; EFSA (European Food Safety Authority), 2010; Regulation (EU) No 488/2014, 2014; Regulation (EU) 2015/1005, 2015). Therefore, considering the values found in the present study, this species is safe for human consumption.

## 6. Conclusions

The concentrations of toxic heavy metals (Cd, Pb, Hg) present in the exoskeleton of *Cronius ruber* are higher than in muscle, and there are significant differences in the concentration of heavy metals with respect to the sex of the specimens and the study areas. The concentration of heavy metals found in the females compared to the males could be because they are in a reproductive phase in which they have greater nutritional needs, and as such, when feeding, they would absorb more of the metals present in the environment. El Pajar concentrations of Cd in muscle tissue and the exoskeleton are $0.003 \pm 0.002$ mg/kg and $0.001 \pm 0.001$ mg/kg, respectively, compared to concentrations in Agaete of $0.003 \pm 0.003$ mg/kg in muscle and $0.004 \pm 0.004$ mg/kg in the exoskeleton.

The significant differences in the study areas suggest that there is pollution in the marine environment. This heavy metal pollution is more than likely due to the tourism in the two locations.

*Cronius ruber* could be a useful bioindicator of Hg, Pb, and Cd pollution and could be used to monitor the pollution of these metals in different geographic areas. It could be a noninvasive monitoring technique if the exoskeleton molts were analyzed.

The concentrations found do not pose any risk to public health, according to the mean values of the estimated daily intake (EDI) for the heavy metals studied and the MOS index. Thanks to this, a catch plan for this crab can be carried out for consumption. If this species were to become a fishing target, the fishermen would catch a greater number of these and, thus, the number of individuals in the marine ecosystem would decrease, so there would not be such a negative effect of this invasive species.

As the intake of this invasive species is fit for consumption, a series of measures could be put in place in order to monitor it. These measures should include a strategy and different actions to prevent this species from damaging the Canary marine ecosystem. On the other hand, collaboration from the different public bodies in preparing and applying rules and regulations is key to mitigating the growth of this species. It is also important to promote research and monitoring in this respect. Finally, and most importantly, the promotion of education and dissemination would help to gain support and encourage responsibility in the fishing community and the rest of the population in this regard.

**Author Contributions:** Conceptualization, T.T.-B., E.L.-B. and A.J.G.; methodology, T.T.-B. and R.T.-P.; software, E.L.-B. and A.J.G.; validation, V.M. and S.P.; investigation, E.L.-B., T.T.-B., S.P. and A.J.G.; resources, A.H.; data curation, C.R.-A.; writing—original draft preparation, T.T.-B., A.J.G. and E.L.-B.; writing—review and editing, A.H. and C.R.-A.; supervision, A.J.G.; funding acquisition, A.H. All authors have read and agreed to the published version of the manuscript.

**Funding:** This research received no external funding.

**Institutional Review Board Statement:** No applicable.

**Informed Consent Statement:** No applicable.

**Data Availability Statement:** No applicable.

**Conflicts of Interest:** The authors declare no conflict of interest.

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
