# Peer review of "Metallic Study of the Invasive Species Cronius ruber—Assessment of Toxic Risk"

_applsci, doi:10.3390/app12073217_

Round 1

Reviewer 1 Report

Review for the paper "Metallic study of the invasive species Cronius ruber. Assessment of toxic risk" by T. Thorne-Bazarra, E. Lozano-Bilbao, R. Triay-Portela, V. Martín, A. Hardisson, C. Rubio, S. Paz, and A. J. Gutiérrez submitted to "Applied Sciences".

General comment.

Living organisms accumulate heavy metals in their tissues. This process needs to be studied especially in animals used for human consumption as there is a tendency to accumulate heavy metals in food webs and top predators. To assess the content of heavy metals such as Cd, Pb, and Hg which are considered as dangerous contaminants in tissues and shells of the invasive crab Cronius ruber, the authors conducted a study at two sites in Gran Canaria (Spain). The authors presented the concentrations of these heavy metals in relation to sampling location and crab sex.

The study may be useful for a wide range of scientists dealing with the assessment of animal health in areas threatened with environmental pollution. The authors have revised the initial draft but some corrections are required.

Recommendations.

The authors well described the problems of environmental pollution and biological invasions but provided very limited information about their study object. They should include more data on the invasion (first occurrence, range expansion), biology (size range, reproduction) and fishery potential (abundance, catch) of Cronius ruber. The section in the results (L 159-161) should be moved in the introduction.

L 98–133. Please, use "Step 1, 2,…" instead of "1 step, 2 step…". Also, check the text. For example, there are two "step 2" (L 98 and 99).

L 128-129. Were these differences significant? The authors should compare the mean values using standard approaches (ANOVA or Kruskal-Wallis test if the data failed tests for normality and homogeneity).

Specific remarks.

L 14. Consider replacing " widely varied diet is" with " widely varied diet are".

L 22. Consider replacing "thus not be harmful to" with "thus not be harmful for".

L 64. Consider replacing "good study organisms to be bioindicators" with "good bioindicators".

L 69. Consider replacing " objective of this study is to know " with " objective of this study is to reveal".

L 72. Consider replacing "For this" with "For this reason".

L 79. Consider replacing "female specimens an" with "female specimens and".

L 94. Consider replacing "formulas for calculate" with "formulas for calculating".

L 110, 130. "Cronius ruber" should be in italics.

L 132. Consider replacing " their respective standard deviation" with " their respective standard deviations".

L 148. Consider replacing " Table 2" with " Table 3".

L 151. Consider replacing " Table 5" with " Table 2".

L 174, 183. Consider replacing " Table 3" with " Table 4".

L 176. Consider replacing "Agaete location" with "the Agaete location".

L 181. Consider replacing " by the PCoA" with " with the PCoA".

L 189, 203. Consider replacing " Table 4" with " Table 5".

L 194. Consider replacing " Table 1" with " Table 2".

L 200. Consider replacing " confirmed in" with " confirmed by".

L 212. Delete "and how the sex factor  influences this".

L 219. Consider replacing " diet’s" with " diets".

L 228. Delete "of crab or other organisms".

L 237. Consider replacing " species of crab" with " crab species".

L 248, 281. Consider replacing " Table 5" with " Table 6".

L 252. Consider replacing "suggests notable problems" with "suggest notable problems".

L 261. Consider replacing " in muscle" with " in the muscle".

L 278. Consider replacing " can be used" with " can also be used".

L 283, 288. Consider replacing " Table 6" with " Table 7".

L 293. Consider replacing " is high, 734 kg" with " is as high as 734 kg".

L 368. " Chiloscyllium plagiosum" should be in italics.

L 376. " Sparisoma cretense" should be in italics.

Author Response

Recommendations.

The authors well described the problems of environmental pollution and biological invasions but provided very limited information about their study object. They should include more data on the invasion (first occurrence, range expansion), biology (size range, reproduction) and fishery potential (abundance, catch) of Cronius ruber. The section in the results (L 159-161) should be moved in the introduction.

  • Added and modified in the manuscript

L 98–133. Please, use "Step 1, 2,…" instead of "1 step, 2 step…". Also, check the text. For example, there are two "step 2" (L 98 and 99).

  • Has been changed in the manuscript

L 128-129. Were these differences significant? The authors should compare the mean values using standard approaches (ANOVA or Kruskal-Wallis test if the data failed tests for normality and homogeneity).

  • Has been changed in the manuscript

Specific remarks.

L 14. Consider replacing " widely varied diet is" with " widely varied diet are".

  • Has been changed in the manuscript

L 22. Consider replacing "thus not be harmful to" with "thus not be harmful for".

  • Has been changed in the manuscript

L 64. Consider replacing "good study organisms to be bioindicators" with "good bioindicators".

  • Has been changed in the manuscript

L 69. Consider replacing " objective of this study is to know " with " objective of this study is to reveal".

  • Has been changed in the manuscript 

L 72. Consider replacing "For this" with "For this reason".

  • Has been changed in the manuscript

L 79. Consider replacing "female specimens an" with "female specimens and".

  • Has been changed in the manuscript

L 94. Consider replacing "formulas for calculate" with "formulas for calculating".

  • Has been changed in the manuscript

L 110, 130. "Cronius ruber" should be in italics.                

  • Has been changed in the manuscript

L 132. Consider replacing " their respective standard deviation" with " their respective standard deviations".

  • Has been changed in the manuscript

L 148. Consider replacing " Table 2" with " Table 3".

  • Has been changed in the manuscript

L 151. Consider replacing " Table 5" with " Table 2".               

  • Has been changed in the manuscript

L 174, 183. Consider replacing " Table 3" with " Table 4".

  • Has been changed in the manuscript

L 176. Consider replacing "Agaete location" with "the Agaete location".

  • Has been changed in the manuscript

L 181. Consider replacing " by the PCoA" with " with the PCoA".

  • Has been changed in the manuscript

L 189, 203. Consider replacing " Table 4" with " Table 5".

  • Has been changed in the manuscript

L 194. Consider replacing " Table 1" with " Table 2".

  • Has been changed in the manuscript

L 200. Consider replacing " confirmed in" with " confirmed by".

  • Has been changed in the manuscript

L 212. Delete "and how the sex factor  influences this".

  • Has been changed in the manuscript

L 219. Consider replacing " diet’s" with " diets".

  • Has been changed in the manuscript

L 228. Delete "of crab or other organisms".

  • Has been changed in the manuscript

L 237. Consider replacing " species of crab" with " crab species".

  • Has been changed in the manuscript

L 248, 281. Consider replacing " Table 5" with " Table 6".

  • We have left Table 6 since we reference that table

L 252. Consider replacing "suggests notable problems" with "suggest notable problems".

  • Has been changed in the manuscript

L 261. Consider replacing " in muscle" with " in the muscle".

  • Has been changed in the manuscript

L 278. Consider replacing " can be used" with " can also be used".

  • Has been changed in the manuscript

L 283, 288. Consider replacing " Table 6" with " Table 7".

  • There is no table 7

L 293. Consider replacing " is high, 734 kg" with " is as high as 734 kg".                    

  • Has been changed in the manuscript

L 368. " Chiloscyllium plagiosum" should be in italics.

  • Has been changed in the manuscript

L 376. " Sparisoma cretense" should be in italics.

  • Has been changed in the manuscript

Reviewer 2 Report

Thank you for submitting your manuscript to the Applied Sciences journal. Generally, the manuscript fits into the scope of the journal, and the structure respects Scientific Best Practice. The manuscript is generally interesting but due to the writing style it is exhausting to read, moreover the methodology could be presented in a more illustrative way. Furthermore, the content needs some revision. In the literature review, it is important that the scientific novelty of the work is established through a critical analysis of related literature. How does this work contribute towards the gaps identified? How does it improve upon previous work? Thus, the main questions of the reviewer are: What is the scientific motivation for the study? Which scientific question shall be answered with this? What is your scientific hypothesis that you wish to answer with the investigation? Putting the scientific motivation will also help you to identify the novelties that characterises a scientific publication. Also the methodology section should be upgraded. I strongly recommend to include a flow chart illustrating the steps of the methodology. Moreover,
what is now numbered as section 3, should b included into the Materials
and Methods section. Figure 1 contains more white than information and
should be upgraded. Regarding the chemical analysis, there should be given
information if the measurements have been done accoding to DIN norma,
and if so, which norms. In the results section is missing the red line. I propose to include
subsections to give the results section a more clear structure. Even
better wuld be to separate a discussion section. Moreover, the
scientific interpretation leaves room for more information. For instance
the bocemical behaviour of the metals should be addressed, means
particularly the respective metal mobility (means particularly the pH-
eH-behaviour that determines the mobility) or non-mobility (means
sorption and bioaccumulation capacity), to support the explanaton on
the different metal contents in bones and muscles. In the conclusions, in addition to summarising the actions taken and
results, please strengthen the explanation of their significance. It
is recommended to use quantitative reasoning comparing with appropriate
benchmarks, especially those stemming from previous work.

Author Response

Thank you for submitting your manuscript to the Applied Sciences journal. Generally, the manuscript fits into the scope of the journal, and the structure respects Scientific Best Practice. The manuscript is generally interesting but due to the writing style it is exhausting to read, moreover the methodology could be presented in a more illustrative way. Furthermore, the content needs some revision. In the literature review, it is important that the scientific novelty of the work is established through a critical analysis of related literature. How does this work contribute towards the gaps identified? How does it improve upon previous work? Thus, the main questions of the reviewer are: What is the scientific motivation for the study? Which scientific question shall be answered with this? What is your scientific hypothesis that you wish to answer with the investigation? Putting the scientific motivation will also help you to identify the novelties that characterises a scientific publication. Also the methodology section should be upgraded. I strongly recommend to include a flow chart illustrating the steps of the methodology. Moreover,what is now numbered as section 3, should b included into the Materials and Methods section. Figure 1 contains more white than information and should be upgraded. Regarding the chemical  analysis, there should be given information if the measurements have been done accoding to DIN norma, and if so, which norms. In the results section is missing the red line. I propose to include subsections to give the results section a more clear structure. Even better wuld be to separate a discussion section. Moreover, the scientific interpretation leaves room for more information. For instance the bocemical behaviour of the metals should be addressed, means particularly the respective metal mobility (means particularly the pH- eH-behaviour that determines the mobility) or non-mobility (means sorption and bioaccumulation capacity), to support the explanaton on the different metal contents in bones and muscles. In the conclusions, in addition to summarising the actions taken and results, please strengthen the explanation of their significance. It is recommended to use quantitative reasoning comparing with appropriate benchmarks, especially those stemming from previous work.

  • The methodology section has been modified and the flowchart added, the introduction and objective part was modified in previous revisions. Figure 1 has been modified and changed, in which it is better shown. We believe that it is for the best understanding of this study that results and discussion should not be divided. The chemical part of the methodology has been modified and more information has been added as well as the conclusions.

Round 2

Reviewer 2 Report

Thank you for providing the revised version. The majority of my comments has been considered in the revison, however the problems with the section Results and discussion not yet. This section is still confusing and difficult to read due to the missing red line. I strongly recommend to improve this section.

Author Response

Review

- Thank you for providing the revised version. The majority of my comments has been considered in the revision, however the problems with the section Results and discussion not yet. This section is still confusing and difficult to read due to the missing red line. I strongly recommend to improve this section.

  • We have separated the results and discussion section. We have also added subsections in the two sections in order to improve the understanding of them.

This manuscript is a resubmission of an earlier submission. The following is a list of the peer review reports and author responses from that submission.

Round 1

Reviewer 1 Report

Review for the paper "Metallic study of the invasive species Cronius ruber. Assessment of toxic risk" by T. Thorne-Bazarra, E. Lozano-Bilbao, R. Triay-Portela, V. Martín, A. Hardisson, C. Rubio, S. Paz, and A. J. Gutiérrez submitted to "Applied Sciences".

General comment.

Living organisms accumulate heavy metals in their tissues. This process needs to be studied especially in animals used for human consumption as there is a tendency to accumulation of heavy metals in food webs and top predators. To assess the content of heavy metals such as Cd, Pb, and Hg which are considered as danger contaminants in tissues and shells of the invasive crab Cronius ruber, the authors conducted a study at two sites in Gran Canaria (Spain). The authors presented the concentrations of these heavy metals in relation to sampling location and crab sex.

The study may be useful for a wide range of scientists dealing with the assessment of animal health in areas threatened with environmental pollution. Some improvements are required to clarify the methods and delete redundant sections. The presentation of the results should be more accurate.

Specific remarks.

Abstract.

Consider replacing "areas with more tourism" with "areas with higher tourism activity".

Introduction, page 2. Consider including the reference after "native species by humans are considered one of the main threats to biodiversity".

Introduction, page 2. Consider replacing "of the present study is to study whether" with " of the present work is to study whether".

Materials and Methods, page 2. The authors should include a map of the study area or coordinates of both sites.

The authors should use equations included in the text instead of Figure 1 presenting these formulae. Each abbreviation should be described for example NOAEL, wd…

Materials and Methods, page 2. Consider replacing "were determined in both specimens." with "were determined in the crabs from both locations".

PERMANOVA requires normal distribution of data. Did the authors check their data for normality and heterogeneity?

Results and Discussion.

The authors stated that they determined “different biometric measurements” but presented no data regarding these measurements. Without these data, it is difficult to evaluate the relevance of the results. The reader should understand how many mature or immature specimens were analyzed.

Moreover, the authors should provide a brief description of the biology of this species to explain the relevance of breading season as a factor affecting the accumulation of metals in females.

The authors should include the minimum and maximum values for their measurements in Table 1. Figures 2 and 4 are redundant and should be deleted. Tables 2, 3 and 4 should be merged into one table. Figures 3, 5, and 6 should be merged into one figure with 3 plates.

Page 4. Consider replacing " The levels of Pb in the exoskeleton is high" with " The level of Pb in the exoskeleton is high".

Conclusion

Page 10. Consider replacing " they would absorb the more of the metals present in" with " they would absorb more of the metals present in".

Page 10. Consider replacing " would help to gain the support" with " would help to gain support".

Author Response

Dear reviewer,
Thank you very much for your considerations. We have chanded and rewriten all of yours indications. 

Reviewer 2 Report

The manuscript entitled “Metallic study of the invasive species Cronius ruber. Assessment of toxic risk” investigates three toxic heavy metals (Cd, Pb and Hg) in the invasive species of Cronius ruber in Canary Islands. The authors found that its high growth rate and widely varied diet are affecting the Canary marine ecosystem. They found significant differences in terms of the location and sex of the specimens, with the highest concentrations being found in areas with more tourism and in the female specimens. They also found that human consumption of this species does not cause any toxic risk. However, the content of this manuscript is important but the overall presentation is suffered to establish the results and conclusion. A major revision is required for considering this manuscript for publication.

Comments:

  • Methodology of atomic absorption spectrometry (GF-AAS) and cold vapor atomic absorption spectrophotometry (CV-AAS) should be included and require explaining step by step.
  • Figure 1 is a formula. It should not be a figure. It will be switched to the text of the methodology section.
  • All figure legends should be elaborated for a better understanding of the results. Also, how the results obtained from each figure are not clear.
  • Results and discussion have been started with the table. This section should be started with the text. The discussion is poorly written, and I would suggest separating the discussion section from the results section.
  • The authors didn’t follow the journal’s guidelines for references input. In the text, they cited authors’ names but in the reference section, they used numbers. Also, most of the references are old and updated references are required for justifying their results.  

Author Response

Dear reviewer, 
Thank you for your revision. We have tried to modify the article attent to your considerations to improve the quality of this.
Best regards

Reviewer 3 Report

Thank you for submission of your manuscript to the Applied Sciences
journal. Generally, the manuscript fits into the scope of the journal,
however there is the need of revision.
In the literature review, it is important that the scientific novelty of the work is established through a critical analysis of related literature. Furthermore, in the introduction must be given a general overview on the subject under investigation, and the motivation why the study is performed as well as the detailed scope of the study. Thus, the main questions of the reviewer are: What is the scientific motivation for the study? Which scientific question(s) shall be answered with this? What is your scientific hypothesis that you wish to answer with the investigation? Putting the scientific motivation will also help you to identify the novelties that characterises a scientific publication. Moreover, the red line in the manuscript is missing. In the current form it is just a collection of of information, but for which purpose
remains unclear. I couldnt undestand why metal oncentrations in
Cronius ruber shall be a problem.

Moreover, I recommend to include a flowchart that illustrates
the steps of the methodology. In the conclusions, in addition to summarising the actions taken and results, please strengthen the explanation of their significance. It is recommended to use quantitative reasoning comparing with appropriate benchmarks, especially those stemming from previous work.
Some information on the uncertainty must be provided.

Author Response

Dear reviewer, 
Thank you very much for your considerations. We have try to response and do all your recomendations.
Kind Regards

Round 2

Reviewer 1 Report

Second Review for the paper "Metallic study of the invasive species Cronius ruber. Assessment of toxic risk" by T. Thorne-Bazarra, E. Lozano-Bilbao, R. Triay-Portela, V. Martín, A. Hardisson, C. Rubio, S. Paz, and A. J. Gutiérrez submitted to "Applied Sciences".

The authors have improved the manuscript but some concerns have still not been considered.

1) The authors should include equations in the text, not in Figure. This requirement is standard for scientific papers.

2) Table 1 should be updated with the maximum and minimum levels as well as the number of individuals in each category. The authors should indicate which kind of error was used here.

3) Page 2. Consider replacing "organism, with fishing it could be decrease the number of these" with "organism, and fishing can decrease its abundance".

4) Page 4. Consider replacing "(Anderson and Braak, 2003)" with "(Anderson and ter Braak, 2003)".

5) Page 4. Consider replacing "areas, we observe" with "areas. We found".

6) Page 5. Consider replacing "depending on the season" with "seasonal variations".

7) Page 11. Source 3. Consider replacing "Braak, C. Ter, " with " ter Braak, C. ".

8) Scientific names should be in italics. See, Pages 11–13, references 5, 6, 12, 18, 24, 32. 37, 40, 41, 42, 46, 54, 55, 57, 58.

9) Both citations and references must be formatted according to Instructions for Authors.

10) The authors should indicate the type of paper (Page 1, Line 1)

Reviewer 2 Report

N/A

Reviewer 3 Report

I reeived the new version which does differ from the previous version only in marginal way. The key issues of my original comments have not been considered.

The current version is still far from having a sufficient scientific quality. The introduction is poor, the novelty is not proven und the description of the methodology is poor. The reader has not any option to proof the provided data. It is even not given the number of samples that was investigated.

As I gave the authors the chance to substantially improve the manuscript and they didnt do this, I disagree to continue with that manuscript.